# The Startling Role of Mismatch Repair in Trinucleotide Repeat Expansions

**DOI:** 10.3390/cells10051019

**Published:** 2021-04-26

**Authors:** Guy-Franck Richard

**Affiliations:** Institut Pasteur, CNRS UMR3525, 25 rue du Docteur Roux, 75015 Paris, France; gfrichar@pasteur.fr; Tel.: +33-1-45-68-84-36

**Keywords:** microsatellites, trinucleotide repeats, mismatch repair, MutS, MutL

## Abstract

Trinucleotide repeats are a peculiar class of microsatellites whose expansions are responsible for approximately 30 human neurological or developmental disorders. The molecular mechanisms responsible for these expansions in humans are not totally understood, but experiments in model systems such as yeast, transgenic mice, and human cells have brought evidence that the mismatch repair machinery is involved in generating these expansions. The present review summarizes, in the first part, the role of mismatch repair in detecting and fixing the DNA strand slippage occurring during microsatellite replication. In the second part, key molecular differences between normal microsatellites and those that show a bias toward expansions are extensively presented. The effect of mismatch repair mutants on microsatellite expansions is detailed in model systems, and in vitro experiments on mismatched DNA substrates are described. Finally, a model presenting the possible roles of the mismatch repair machinery in microsatellite expansions is proposed.

## 1. Microsatellites and Mismatch Repair

Microsatellites are short sequence repeats (SSR), whose base motif is 1–9 bp repeated in tandem. They have been encountered in all genomes sequenced so far, although they are less frequent in prokaryotes. In eukaryotes, several dozen to several hundred microsatellites are found per megabase of nuclear DNA [1]. In our genome itself, they account for 3% of the total sequence, similar to the amount of coding sequence [2]. Their tandemly repeated nature makes them prone to small length changes by slippage, occurring during DNA synthesis associated with S-phase replication, gene conversion, or DNA repair [1].

Almost 30 years ago, it was shown in the budding yeast *Saccharomyces cerevisiae*, that the rate of instability of a (GT)_30_ dinucleotide repeat was 10^−4^ per cell division [3], orders of magnitude higher than single nucleotide substitutions [4]. The second important discovery of this yeast study was that no effect of a microsatellite’s orientation on its stability could be seen. Repeat length changes happened with similar frequencies when the GT sequence was on the lagging-strand template or on the leading-strand template, showing that no preferential newly synthesized strand was more prone to slippage. Shortly thereafter, it was discovered that the mismatch repair machinery (Hereafter abbreviated “MMR”) was directly responsible for fixing replication errors made during replication slippage within microsatellites. Using an elegant experimental system, relying on an *URA3* reporter gene in which a (GT)_33_ microsatellite was integrated, Thomas Petes and collaborators showed that inactivation of *MSH2*, *MLH1,* or *PMS1* increased the rate of instability in the GT tract by several hundred-fold [5]. Microsatellite sequencing showed that most of the length changes involved additions or subtractions of one dinucleotide (2 bp). Later on, it was proven that mononucleotide repeats were also destabilized in mismatch-repair deficient yeast strains. The +1 or −1 bp instability of poly-A, poly-C, poly-G, or poly-T microsatellites increased by several thousand-fold in the absence of functional mismatch repair [6,7,8]. Additional experiments showed that microsatellite instability during replication increased with increasing tract length [9] and decreased when the repeat tract was interrupted by variant motifs [10]. Microsatellite instability was also observed with tetra- to octa-nucleotide repeats, although the effect of the mismatch repair machinery was much stronger on shorter motifs [11]. Similar observations were made in the fission yeast *Schizosaccharomyces pombe* [12].

At the same time these important discoveries were being made in yeast, it was found that hereditary non-polyposis colon cancer (HNPCC) was associated with a mutation of the human MSH2 gene [13]. In this type of cancer, microsatellite instability is increased by several orders of magnitude compared with non-cancerous cells. Several genes whose function are related to cell proliferation were subsequently found to be mutated in such cancer lines. The type II transforming growth factor β [14], IGFIIR, the insulin-like growth factor II receptor [15], and BAX, regulating apoptosis [16] were all found to contain somatic mutations in microsatellites, leading to repeat length changes in mono- or dinucleotide repeats. As was observed in budding yeast, most mutations were additions or deletions of one repeated motif.

## 2. Trinucleotide Repeat Expansions and Human Disorders

Concomitantly, the first of a rapidly growing family of human disorders was found to be linked to the expansion of trinucleotide repeats, a class of microsatellites with very peculiar properties. Fragile X syndrome, the most common cause of hereditary mental retardation, and characterized by chromosomal fragility, is triggered by an expansion of a CGG trinucleotide repeat in the 5’ UTR of the FMR1 gene [17]. Huntington’s chorea, a dramatic neurodegenerative disorder is due to an expansion of a CAG triplet repeat within the first exon of the gene encoding the huntingtin protein [18]; myotonic dystrophy type 1 (also called Steinert disease or DM1) is induced by the expansion of a CTG trinucleotide repeat in the 3’ UTR of the DMPK gene [19]; and Friedreich’s ataxia, a recessive neurological disorder characterized by a defect in iron metabolism, is provoked by the expansion of a GAA repeat in the first intron of the FRDA gene [20]. Shortly after, it would be discovered that other microsatellites besides trinucleotide repeats could be found expanded in such disorders (in the present review, “trinucleotide repeat” or “triplet repeat” will be used indiscriminately for any microsatellite prone to expansion linked to a human disorder). This is the case with myotonic dystrophy type 2, induced by the expansion of a CCTG repeat [21], and amyotrophic lateral sclerosis (ALS) due to the expansion of a GGGGCC hexanucleotide repeat [22]. At the present time, microsatellite expansions are responsible for approximately thirty neurological or developmental disorders [23,24]. Several molecular mechanisms, including S-phase DNA replication slippage, homologous recombination, and DNA repair, have been shown to modulate trinucleotide repeat expansions and contractions [25,26,27,28,29]. These DNA sequences exhibit length polymorphism in the non-affected population, a property shared by other microsatellites. However, there are four specific features that distinguished them from other, non-expandable, microsatellites:All of them have been shown to form stable secondary structures in vitro, and this unusual feature was proposed to trigger the expansion process [30,31]. CAG and CTG trinucleotide repeats are able to form imperfect hairpins, CGG triplet repeats fold into hairpins or G-quadruplex, and GAA repeats have the ability to form triple helices, containing both Watson–Crick and Hoogsteen bonds. At the present time, the possibility that these secondary structures also form in living cells is still a matter of debate [32].Trinucleotide repeats exhibit a clear bias toward expansions, at each generation. There are, however, reported cases of rare contractions suppressing the symptoms of the pathology [33,34].Unlike other microsatellites that generally increase or decrease their length by one repeated motif, trinucleotide repeat expansions behave differently. They expand from a few triplets at a time, like in Huntington’s disease, to hundreds or even thousands of triplets in one single generation, like in myotonic dystrophies 1 and 2. Expansions (or contractions) of more than one repeat unit may happen in other microsatellites, but they are rare and never reach the length alterations seen with trinucleotide repeats.The instability of a given trinucleotide repeat is highly dependent on its orientation during replication. This was first demonstrated in *Escherichia coli* [35] and soon after confirmed in yeast [36,37]. When a CAG/CTG repeat tract was replicated in such a way that the CTG sequence was on the lagging-strand template, frequent contractions were observed and almost never expansions. However, when the CAG sequence was on the lagging-strand template, the general instability was reduced and some expansions were visible (although contractions remained predominant). This was explained by a more frequent formation of secondary structures on the lagging-strand template and by the observation that CTG hairpins are more stable than CAG hairpins [38].

Despite these clear-cut differences between trinucleotide repeats and other non-expandable microsatellites, early experiments designed to determine the mechanisms by which these expansions occurred, naturally turned to the usual suspect: the mismatch repair machinery.

## 3. A Tale of Yeast, Mice, and Men

### 3.1. Early Yeast Experiments

Experiments designed to test the effect of mismatch repair deficiency on trinucleotide repeats were first carried out in yeast, in which eukaryotic MMR genes were first identified and studied. Trinucleotide repeats of different lengths were integrated into a chromosome, in yeast cells deficient in mismatch repair genes. Repeat tract length was assessed by PCR run on sequencing gels in order to detect changes as small as one triplet. Unsurprisingly, inactivation of *MSH2* or *PMS1* increased the frequency of small changes (±1–2 triplets) [39], similarly to what was already described for other microsatellites. However, the frequency of larger expansions and contractions was apparently unchanged.

An elegant genetic assay was designed by Bob Lahue in *S. cerevisiae*, taking advantage of the fact that the *S. pombe ADH1* promoter exhibits specific spacing requirements to function in budding yeast. A CTG trinucleotide repeat was integrated in this promoter and the *ADE8* gene was used as a reporter in a colony color assay. When the repeat was long enough, the promoter was off, and the yeast cells were white. When repeat contraction occurred, the promoter was turned on, and the cells became red [40]. This assay was subsequently adapted to score expansions with other reporter genes besides *ADE8*. It presented the advantage of scoring only large expansions or contractions. Therefore, all small additions or subtractions of one or two triplets would remain undetected. The effect of MMR mutants on CTG trinucleotide instability was addressed using this experimental system. Remarkably, it showed that deletion of *MSH2* or *MSH3* dramatically decreased expansion rates in both CAG and CTG orientations, using different repeat tract lengths (Table 1) [41,42,43]. Interestingly, *MSH6* had a small opposite effect, its inactivation slightly increased expansion rates in both orientations [42]. The Msh2 and Msh6 proteins form the MutSα heterodimeric complex, involved in detecting and fixing single base mismatches and small insertions or deletions, whereas Msh2 and Msh3 form the MutSβ complex, whose function is to repair larger indels [44,45]. Therefore, these results suggested that large loops were leading to expansions by a mechanism dependent on a functional MutSβ complex. Chromatin immunoprecipitation of Msh2 showed an enrichment of this protein at a long CAG/CTG repeat in yeast. This enrichment was lost in a *msh3*Δ mutant in both orientations, and in a *msh6*Δ mutant, only in the CAG orientation [46]. This suggested different roles for both MutS complexes at trinucleotide repeats.

Interestingly, the effect of MMR mutations on large repeat contractions were, most of the time, not significant in most experimental systems [39,40,47], proving that large CAG/CTG trinucleotide repeat expansions and contractions happened in yeast by distinct mechanisms.

The expansion frequency of a (GAA)_100_ sequence integrated in a yeast chromosome increases by two-fold in a *msh2*Δ mutant [48], and the frequency of arm loss following chromosomal breakage decreases by 30-fold in the same mutant [49]. This rather modest increase, as compared to what was observed with CAG/CTG repeats, suggests that the MMR may play a significantly different role in GAA repeat metabolism.

### 3.2. Trinucleotide Repeat Expansions in Mice Are Mismatch-Repair Dependent

The effect of mismatch repair on trinucleotide repeat expansions was also studied in transgenic mice. A long CTG trinucleotide repeat from a DM1 patient integrated into the mouse genome exhibited both intergenerational and somatic instability [50], and in some cases very large expansions, similar to those observed in the human population [51]. The trinucleotide repeat was replicated from a downstream origin, in such a way that the CAG sequence was on the lagging-strand template, the expansion-prone orientation [52]. Inactivation of both copies of *Msh2* in animals carrying an expanded human allele of the Huntington’s disease gene led to a clear reduction of the instability in all tissues studied, with a marked decrease in expansion size [53]. In a DM1 mouse model, expansions were suppressed in the sperm, quadriceps, and cerebellum of a *Msh2* mutant, [54]. *Msh2*-dependent expansions occurred in spermatogonia as early as at seven weeks of age and continued throughout life [55]. It was subsequently showed that *Msh3* inactivation led to a very similar phenotype, both in maternal and paternal transmissions, whereas *Msh6* deletion decreased expansions only during maternal transmissions [56], suggesting slightly different roles for both MutS complexes, a finding reminiscent of what was previously observed in yeast. Intergenerational instability depends on *Msh2* ATPase activity, its inactivation by a point mutation leading to the same phenotype as a null mutant [57]. It was suggested that large CAG/CTG trinucleotide repeat hairpins could bind MMR proteins in an inactive conformation, “hijacking” the repair activity, and ultimately leading to expansions. However, experiments using HeLa cell extract did not confirm this hypothesis, since both MutSα and MutSβ competent extracts were shown to efficiently repair CAG and CTG hairpins [58].

Interestingly, *Msh3* was also shown to be driving expansions in a mouse model for Huntington’s disease. In a mice background in which *Msh3* is expressed at a low level, expansions are decreased similarly to a *Msh2* null animal [59]. It is known that MMR protein levels vary between mouse tissues, but *Msh3* is often more abundant than *Msh6* [60]. This may partly explain the tissue-specific instability of CAG/CTG trinucleotide repeats. *Msh3*, as well as *Mlh1*, were also shown to be specifically involved in expansions in a transgenic mice model for Huntington’s disease [61]. MLH1 together with PMS2, form the MutLα heterodimeric complex, which binds to both MutSα and MutSβ to trigger mismatch repair activity. It is the main MutL player. In addition, MutLβ is made of MLH1 and PMS1 proteins, and its function is not fully understood. Finally, MLH1 and MLH3 assemble into MutLγ, whose primary function is in meiotic recombination. It is also a MutLα backup [44]. In a DM1 mouse model, inactivation of *Pms2*, also decreased the rate of expansions [62], proving that not only MutSβ, but also the MutLα complex (at least), were responsible for inducing CTG repeat expansions [63].

Besides CAG/CTG repeats, which have been the most extensively studied, transgenic mice for CGG/CCG repeats responsible for X fragile syndromes were also constructed [64]. In mice in which both *Msh2* alleles were inactivated, expansions were decreased both during paternal and maternal transmissions, although the effect was more marked during the former [65]. When *Msh3* was inactivated, expansions were suppressed while contractions increased, in both types of transmission [66]. In mice deficient for *Pms2*, *Pms1* or *Mlh3*, expansions were suppressed, showing that all three MutL complexes were involved in the expansion mechanism [67,68].

The third trinucleotide repeat that has been studied in mice is the GAA/TTC repeat tract, responsible for Friedreich’s ataxia. In a mice model, inactivation of both *Msh2* alleles led to an increase of GAA repeat contractions, whereas expansions were unchanged. The same phenotype was observed in a *Msh3* mutant background, with an increase in contractions and no apparent effect on expansions. In *Msh6* null mice, both expansions and contractions were increased, whereas in *Pms2* mutant animals, expansions were largely increased while contractions were decreased [69]. These striking differences with CAG/CTG repeat stability in mice deficient for MMR proteins is reminiscent of what was observed in yeast cells for the same repeat tracts and strongly suggests that the mismatch repair machinery is playing different roles in GAA/TTC repeats compared to CAG/CTG repeats (Table 2).

### 3.3. Genetic Drivers of Trinucleotide Repeat Expansions in Humans

Early on, the question arose of the role of mismatch repair in trinucleotide repeat expansions in humans. A panel of Huntington’s disease patients was screened for mutations at eight other microsatellite loci known to be unstable in colorectal cancer, but little or no instability was detected at any of these loci. Conversely, the HD and SCA1 loci, both containing a CAG trinucleotide repeat, were highly unstable in colorectal cancer cell lines, like any other microsatellite [70]. This suggested that MMR had a stabilizing effect on the CAG repeat tract, as expected, but also that an additional mechanism was contributing to HD instability.

Most of the subsequent efforts have focused on designing transgenic mice that could serve as models for the most common trinucleotide repeat disorders. The effects of mismatch repair mutations were tested in these mice, as hereabove described. This approach was somewhat surprising, since there was no report of trinucleotide repeat expansion disorders in mice, and it was unknown whether mechanisms suspected in humans would be similar to those identified in animals. There was one study reporting the germline and somatic instability of a GGCA tetranucleotide repeat in mice, but not associated with a bias toward expansions [71]. Therefore, undertaking experiments in mice was betting that results obtained in animals would be relevant to humans.

Genome wide association studies (GWAS) rely on the use of known genetic polymorphisms in order to identify those linked to a particular disease or phenotypic trait. Former linkage analysis studies used microsatellite markers, especially CA dinucleotide repeats, which are the most common microsatellites in the human genome [72]. More recent linkage studies have depended on SNPs, which are much more frequent, although somewhat less informative than microsatellites. A large GWAS study was built using several cohorts of patients affected with Huntington’s disease and genotyped by SNP arrays. This study identified two SNPs, on chromosomes 8 and 15, that were significantly associated with the “residual age of onset” of the disease (defined as the difference between the age at which symptoms are observed compared to the age predicted, based on the trinucleotide repeat length), as well as three others, on chromosomes 3, 5, and 21, for which association was suspected, but just below significance. Identification of the genes in these regions found they belonged to three metabolic categories. DNA repair was particularly well represented, with hits in *MLH1* and other mismatch repair genes, as well as in *FAN1*, a gene encoding a nuclease involved in processing inter-strand DNA links. The two other categories were genes involved in oxydo-reductase activity or in mitochondrial and peroxisomal metabolisms [73]. Another GWAS study, using different HD cohorts, led to the identification of several MMR genes, including *MSH3*, *PMS2,* and *MLH1* [74]. Later on, it was found that the *MSH3* gene contains a 9 bp tandem repeat in its coding sequence, exhibiting both length and sequence polymorphisms. One of its shortest alleles was associated with HD and DM1 phenotypes, suggesting that both disorders may have a common genetic origin [75]. Another *MSH3* allele was found associated with DM1 in a Costa Rican cohort of 199 individuals [76]. Interestingly, knockdown of *MLH1* or *MLH3* using shRNA in a human cell model of Friedreich’s ataxia led to a significant reduction of GAA/TTC trinucleotide repeat expansions, but knockdown of *PMS2* led to a small increase, reminiscent of what was observed in transgenic mice [77]. It must be noted that GAA/TTC repeats show a bias toward expansions in induced pluripotent stem cells (iPSC), in which both MutSα and MutSβ complexes are abundant as compared to fibroblasts or neurospheres [78].

In conclusion, experiments in yeast, mice, and human cells, as well as GWAS data, all point to a role of MMR proteins in trinucleotide repeat expansions by a mechanism that seems to be unrelated to classical strand slippage during the replication of non-expandable microsatellites [79].

## 4. Slipped-Stranded DNA, Trinucleotide Repeats, and Mismatch Repair Proteins

One of the very first experiments that suggested that MMR proteins exhibit unusual interactions with CAG/CTG trinucleotide repeats was performed by Christopher Pearson, using purified human MSH2 protein and slipped-stranded DNA structures. Plasmids containing 30 or 50 CAG/CTG triplets from the DM1 locus were denatured, labeled, mixed, and renatured in equimolar conditions in order to form different homo- and heteroduplex DNA species. Linear homoduplexes containing either 30 or 50 triplets did not bind MSH2, whereas slipped-stranded duplexes containing 30 or 50 triplets showed a strong interaction with MSH2, by band-shift assays. Interestingly, the shifted signal was stronger with an excess of CAG compared to CTG, indicating that CAG triplets were bound more efficiently; but the shift was higher with an excess of CTG, suggesting that maybe more proteins were bound to CTG triplets [80]. These results were already pointing to there being subtle differences in CAG/CTG instability, which depends on which strand (CAG or CTG) secondary structures formed. Later experiments using a similar setup showed that nicked heteroduplex slipped-stranded substrates were repaired independently of MMR proteins. This suggested that the mismatch repair machinery may play a role in an earlier step that leads to the formation of such substrates, but not in their subsequent repair [81]. Additional experiments refined these observations, by showing that short slipped-stranded loops were efficiently repaired by MutSβ, and to a lesser extent by MutSα, whereas multiple small loops or large loops could not be fixed by the mismatch repair machinery [82]. Repair of small-loops was also shown to be dependent on MLH1 and PMS2 [83]. These results were confirmed by independent experiments using purified human proteins on synthetic DNA substrates. The authors showed that CAG or CTG loops of fewer than four triplets needed MutSβ and MutLα to be efficiently repaired, whereas loops containing at least four triplets were resistant to repair. In this experimental system MutSα was unable to repair any of the loops tested [84].

Purified MSH2 and MSH3 human proteins were incubated with different substrates containing a small loop or larger CAG hairpins. Intriguingly, ATP hydrolysis by MutSβ was inhibited by CAG hairpins, this inhibition increasing with the number of A•A mismatches on the hairpin. The authors suggested that the CAG secondary structure bound the MSH2–MSH3 complex in an inactive form, compromising its efficient repair [85]. However, an independent study did not find any inhibitory effect of CAG hairpins on MutSβ activity, and this is therefore still an open question [58].

All the above in vitro experiments used slipped-stranded substrates assembled from denatured plasmidic DNA or synthetic sequences. If such structures exist in living cells, it is predicted that heteroduplex DNA will segregate after replication as two alleles of different lengths, giving rise to sectored colonies (Figure 1). This was observed in a cell model, in which mixed colonies were more abundant in a MutSβ deficient background [86]. Interestingly, in a yeast model in which sectored colonies occur frequently in the CTG orientation, overexpression of *MSH2* increased their frequency by 10-fold, in a *MSH3* and *MSH6*-dependent manner. A point mutation in the *MSH2* Walker B motif (Glu768 -> Ala768) that does not affect heterodimerization nor binding, but dramatically decreases ATPase activity, resulted in a partial suppression of sectored colonies [46]. This shows that in yeast the intracellular balance of both MutS complexes is crucial for keeping CTG heteroduplexes at a low level.

## 5. Conclusions and Perspectives

In the light of observations made in model organisms and GWAS analyses, it is obvious that the mismatch repair machinery is involved in trinucleotide repeat instability. However, some important differences with non-expandable microsatellites are noteworthy: (i) a strong bias toward expansions compared to contractions; (ii) the critical role of MutSβ compared to MutSα, suggesting that large mismatches are the rule; and (iii) the frequent occurrence of mixed progeny, suggesting that hairpin-containing heteroduplex DNA frequently escapes repair. Present models propose that expansions occur if the old strand is repaired using the newly synthesized strand as a template, generating a small expansion whose size is equal to the length of the hairpin (Figure 1) [79,87]. Conversely, if the new strand is fixed using the old one as a template, no size change occurs. In support of this mechanism, in vitro experiments have shown that MutLγ induces a nick in the DNA strand that does not contain a short CTG loop, forcing repair to use the loop-containing strand as a template, eventually leading to repeat expansion [88]. In addition, the fragility of a (CAG)_70_ repeat tract was decreased in yeast in *mlh1*Δ and *mlh3*Δ strains, as well as in a *mlh3*-D523N mutant that abolishes MulLγ endonuclease activity. This suggests that MutLγ is responsible for making nicks in CAG/CTG hairpins, increasing chromosomal fragility [89].

Finally, if the mismatch escapes repair by the MMR, the heteroduplex will persist until the next cell division, during which it will segregate into different cells, giving rise to a sectored colony in experimental systems in which such events can be observed. It is likely that this mechanism occurs several times in the same cell line, leading to successive small expansions of the trinucleotide repeat tract. It is also possible that after reaching a given length threshold, another mechanism triggers large expansions such as those observed in several human disorders that are not seen with regular microsatellites. Note that this model may explain results obtained with CAG/CTG and CCG/CGG trinucleotide repeats. The different observations made with GAA/TTC repeats are clearly not totally compatible with this model. This might be due to the very distinct nature of the secondary structures formed by these triplet repeats [32].

It would be futile to conclude this review without a reminder of the crucial discovery made by Miroslav Radman, more than 30 years ago, on the role of the mismatch repair machinery during homologous recombination. The conjugation between *Escherichia coli* and *Salmonella typhimurium* is normally very inefficient, presumably because sequence identity between these two species is too low (~80%). However, when *S. typhimurium* strains with MMR gene defects were used as recipients for conjugation, a 1000-fold increase in RecA-dependent recombination was observed, proving that mismatch repair was inhibiting interspecific homeologous recombination [90]. The role of mismatch repair in trinucleotide repeat instability has been addressed in *E. coli*, by several independent laboratories. Despite disparate observations, a few clear conclusions may be drawn from these studies. Mutations in *mutS* destabilize CAG/CTG trinucleotide repeats, mainly by increasing the frequency of +1 or −1 triplet mutations [91,92]. In addition, large repeat contractions were less frequent in such mutants [91,93], opposite from what was found in other organisms. This discrepancy may have been because of the use of plasmid-borne repeats or due to subtle differences between bacteria and eukaryotes.

Gene conversion associated with homologous recombination has been definitely linked with frequent expansions and contractions of CAG/CTG trinucleotide repeats in *S. cerevisiae*. These frequent length changes depend on the Mre11–Rad50–Xrs2 complex [47,94], occur during both meiotic and mitotic recombination [95], and do not involve unequal crossovers [96]. The precise role of the mismatch repair machinery in this highly regulated process has not yet been addressed, but it might help throw light on the amazingly complex molecular processes regulating trinucleotide repeat instability in eukaryotes.

## Figures and Tables

**Figure 1 cells-10-01019-f001:**
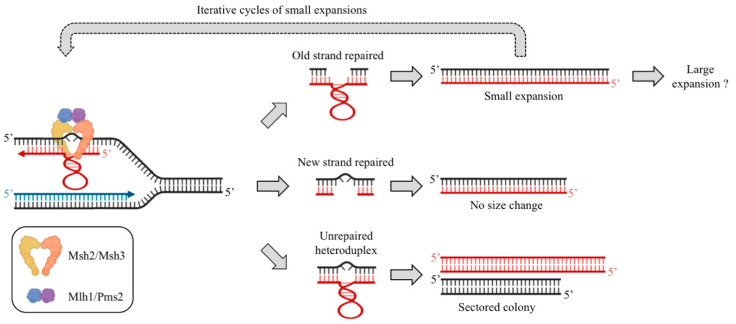
Possible roles of mismatch repair in trinucleotide repeat instability. Slippage during CAG/CTG replication on the newly synthesized lagging strand generates heteroduplex DNA. This mismatch is recognized by the MutSβ complex (and to a lesser extent by MutSα) and by MutLα (and probably other MutL complexes too). If the old strand is fixed using the newly synthesized strand as a template, a small repeat expansion will occur (Top). If the hairpin is removed, and the newly synthesized strand is fixed using the old strand as a template, no size change occurs. (Middle). If the heteroduplex escapes repair by the MMR, segregation of both strands during the following cell cycle will lead to a mixed progeny (sectored colony). Note that several rounds of small expansions may occur during somatic cell divisions. In addition, above a certain length threshold, a large expansion may occur, by a yet unknown mechanism, which may involve DNA repair and/or recombination.

**Table 1 cells-10-01019-t001:** Trinucleotide repeat expansions per cell division in yeast mismatch repair mutants.

Genotype	Nbr of Triplets	CAG Orientation	CTG Orientation	Reference
WT	25	5 × 10^−7^	1 × 10^−5^	[41]
*msh2*Δ	25	3 × 10^−8^ **(****↓17×)**	9 × 10^−6^ **(=)**	
WT	25	1 × 10^−6^	3 × 10^−5^	[42]
*msh3*Δ	25	2.4 × 10^−7^ **(↓5×)**	1 × 10^−6^ **(↓30×)**	
*msh6*Δ	25	2.4 × 10^−6^ **(↑2×)**	1.5 × 10^−4^ **(↑5×)**	
WT	47–59	4.2–7.8 × 10^−1^	ND	[43]
*msh3*Δ	47–59	1.7–5 × 10^−2^ **(****↓15–25×)** ^(1)^	ND	
WT	30–34	ND	4.3–4.7 × 10^−1^	
*msh3*Δ	30–34	ND	0.8–1.2 × 10^−1^ **(****↓4–5×)** ^(1)^	

^(1)^ Depending on whether expansions are computed after 7 or 14 days.

**Table 2 cells-10-01019-t002:** Effect of mismatch repair mutations on trinucleotide repeat instability, compared to wild-type transgenic mice.

Parental Genotype	CAG/CTG Expansions	CAG/CTG Contractions	Transmission	Reference
*msh2* ^−/−^	**↓**	**↑**	unspecified	[54]
*msh3* ^−/−^	**↓**	**↑**	♂ and ♀	[56]
*msh6* ^−/−^	**=**	**=**	♂	[56]
*msh6* ^−/−^	**↓**	**↑**	♀	[56]
*mlh1* ^−/−^	**↓**	**=**	unspecified	[61]
*pms2* ^−/−^	**↓**	**↑**	unspecified	[63]
*mlh3* ^−/−^	**↓**	**=**	unspecified	[61]
**Parental genotype**	**CGG/CCG expansions**	**CGG/CCG contractions**	**Transmission**	**Reference**
*msh2* ^−/−^	**↓**	**↑**	♂ and ♀	[65]
*msh3* ^−/−^	**↓**	**↑**	♂ and ♀	[66]
*pms2* ^−/−^	**↓**	**↑**	unspecified	[68]
*pms1* ^−/−^	**↓**	**=**	unspecified	[68]
*mlh3* ^−/−^	**↓**	**↑**	unspecified	[68]
**Parental genotype**	**GAA/TTC expansions**	**GAA/TTC contractions**	**Transmission**	**Reference**
*msh2* ^−/−^	**=**	**↑**	unspecified	[69]
*msh3* ^−/−^	**=**	**↑**	unspecified	[69]
*msh6* ^−/−^	**↑**	**=**	unspecified	[69]
*pms2* ^−/−^	**↑**	**↓**	unspecified	[69]

## Data Availability

No new data were created or analyzed in this study. Data sharing is not applicable to this article.

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
