# Peer review of "The Startling Role of Mismatch Repair in Trinucleotide Repeat Expansions"

_cells, 2021, doi:10.3390/cells10051019_

Round 1
Reviewer 1 Report
Manuscript Number: Cells-1184624 “ The startling role of mismatch repair in trinucleotide repeat expansions" by Guy-Franck Richard.
The review by Guy-Franck Richard analyzes the current status of literature on the role of the Mismatch Repair (MMR) in the maintenance of trinucleotide repeats. This is a thorough and accurate review of in vivo and in vitro activities of the complex. The author methodically describes what is known about the activities of the MMR proteins in the maintenance of trinucleotide repeats in yeast, mice, and human cells. Special attention the author paid to in vitro interactions between secondary structures formed by trinucleotide repats and MMR proteins. The author also presents a feasible model of TNR instability induced by MMR. In general, the reader gets a complete and up-to-date picture of the role of MMR in TNR instability.
There are several minor comments and suggestions outlined below:
- In Abstract (line 8) and in line 73, the author gives a very specific number (30) for the disease caused by the instability of microsatellites. As far as know, at least 37 diseases are currently known to arise due to instability of microsatellites. I would rather change to either „approximately 30“ or to find out about the exact numbers of Microsatellite and TNR-mediated diseases at the moment.
- Deficiency in MMR prevents GAA/TTC expansions in cultured iPS cells. This reference is missing:
Du, J., Campau, E., Soragni, E., Ku, S., Puckett, J.W., Dervan, P.B., and Gottesfeld, J.M. (2012). Role of mismatch repair enzymes in GAA.TTC triplet-repeat expansion in Friedreich ataxia induced pluripotent stem cells. The Journal of biological chemistry 287, 29861-29872.
- Considering the model, it is worth mentioning of this recent study:
„Human MutLγ, the MLH1-MLH3 heterodimer, is an endonuclease that promotes DNA expansion.“ Kadyrova LY, Gujar V, Burdett V, Modrich PL, Kadyrov FA. Proc Natl Acad Sci U S A. 2020 Feb 18;117(7):3535-3542.
Author Response
I have addressed all the comments of reviewer #1 and added the two references requested. Text modifications are in red.

Reviewer 2 Report
See attached file.

Author Response
I have added the three references requested by reviewer #2 (one of those was common to both reviewers). Concerning the many grammar and syntax modifications suggested by this reviewer, I have fixed all of them, except when the suggestion was inappropriate. The parts that were left unchanged are:
- Page 3, line 93: Change “reach length alterations seen” to “reach the lengths seen” would be inappropriate since some microsatellites reach lengths that are similar to those of expanded trinucleotide repeats. So, the length alterations are unusual, not the length themselves.
28, 29, 30. Mouse gene names are written Msh2, not MSH2 like in yeast or humans. These names were italicized but not made upper case, as requested by the reviewer. However, mouse protein names were made upper case.
- Page 6, line 226: Change “three metabolic categories” to “three metabolic”. This was left unchanged, since DNA metabolism is part of the cellular metabolism.
All modifications are in red in the revised text.
